# One-volt-driven superfast polymer actuators based on single-ion conductors

Onnuri Kim[1], Hoon Kim[1], U. Hyeok Choi[2,3] & Moon Jeong Park[1]

The key challenges in the advancement of actuator technologies related to artificial muscles include fast-response time, low operation voltages and durability. Although several researchers have tackled these challenges over the last few decades, no breakthrough has been made. Here we describe a platform for the development of soft actuators that moves a few millimetres under 1 V in air, with a superfast response time of tens of milliseconds. An essential component of this actuator is the single-ion-conducting polymers that contain well-defined ionic domains through the introduction of zwitterions; this achieved an exceptionally high dielectric constant of 76 and a 300-fold enhancement in ionic conductivity. Moreover, the actuator demonstrated long-term durability, with negligible changes in the actuator stroke over 20,000 cycles in air. Owing to its low-power consumption (only 4 mW), we believe that this actuator could pave the way for cutting-edge biomimetic technologies in the future.

[1] Department of Chemistry, Pohang University of Science and Technology (POSTECH), Pohang 790-784, Korea. [2] Functional Composites Department, Korea Institute of Materials Science, Changwon 642-831, Korea. [3] Department of Polymer Engineering, Pukyong National University, Busan 608-739, Korea. Correspondence and requests for materials should be addressed to M.J.P. (email: moonpark@postech.ac.kr).

There is a growing demand for low-voltage-driven electromechanical transducers because of their wide use in emerging fields such as soft robotics[1,2] and biomedicine[3,4]. A promising candidate is the ionic polymer actuators, which is capable of large displacement under low operation voltages of only a few volts[5,6]. The unique salient features of ionic polymer actuators include flexibility, ease in manufacturing, low cost and lightweight. Extensive efforts have been made to improve the performance of ionic polymer actuators over the last decade through the discovery of new ionic polymers[5,7–9] and various electrode materials[10,11]. In particular, the introduction of ionic liquids into the actuators has shown great potential for achieving large bending strains with a long-cycle life[5], as ionic liquids offer advantages of a wide window of electrochemical stability and high ionic conductivity[12].

Tailoring the intermolecular interactions between the polymer matrix and the embedded ionic liquid appeared to be vital for controlling the actuation properties[13–15]. The use of a block copolymer matrix that causes microphase separation to form periodic nanostructures is also considered a potential method for developing high-performance ionic polymer actuators[8,14–16], ascribed to the creation of less tortuous ion-conduction pathways[17]. However, the enhancement of the response time of ionic polymer actuators, which reportedly ranges from several seconds to tens of seconds, remains a major challenge[8,16]. The slow response time is attributed to the depletion of cations and anions near the electrode surfaces under an electric field[12,18]. The asymmetric diffusion of cations and anions is also a major impediment to the realization of durable actuators as it causes uncontrolled back relaxation behaviour[19].

These issues may be solved if one of the ions is immobilized to the polymer matrices to form so-called single-ion conductors, thus alleviating the polarization concerns. However, the development of high-performance actuators does not appear to be feasible at present, because such actuators have large deficiencies in their displacements compared with the conventional ionic polymer actuators wherein both the cation and anion are moving-as this approach significantly reduces the ion diffusivity[19–21]. Moreover, degree of ion dissociation in single-ion-conducting polymers is limited because of the intrinsically low dielectric constant of the polymer chains neighbouring the ion[22], which impedes the lowering of the operation voltage of the actuators. If the ionic polymer actuators can respond quickly-in milliseconds like piezoelectric[23] and dielectric actuators[24]—with a small battery (<1.5 V) and without sacrificing its bending strain—it could present a chapter in the development of artificial muscles.

In the present study, we describe a platform for the development of high-performance ionic polymer actuators, which could be used to achieve a superfast response time under low-voltage operating conditions. The actuators comprise single-ion-conducting block copolymers that exhibited a response time of tens of milliseconds and millimetre-scale displacement under 1 V in air, which has not been achieved by any actuators thus far.

## Results

### Cation-conducting block copolymers comprising zwitterions.
Cation-conducting polymers were prepared from sulfonated polymers by doping with imidazole (Im). We designed the polymers to have self-assembled structures by covalently linking sulfonated polystyrene (PSS) and polymethylbutylene (PMB), that is, PSS-b-PMB, which enables ions to be confined into the PSS phases without affecting the mechanical integrity of the

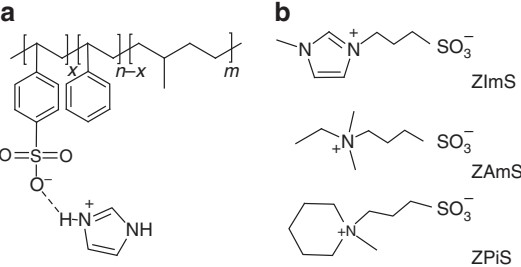

**Figure 1 | Molecular structures.** (**a**) Molecular structure of imidazole-doped PSS-b-PMB block copolymer and (**b**) molecular structures of ZImS, ZAmS and ZPiS.

ionophobic PMB domains. Figure 1a shows the molecular structure of the Im-doped PSS-b-PMB block copolymers. The degree of polymerization ($N = n + m$) of PSS-b-PMB block copolymers was varied from $N = 145$ to 598 in order to optimize the mechanical strength of the polymer layer. Moreover, the sulfonation level (SL, $x/n$) of the PSS chains in PSS-b-PMB was adjusted from 20 to 75 mol%, thereby controlling ionic conductivity. For brevity, we will only discuss the representative data obtained from a single PSS-b-PMB with $n = 153$, $m = 313$ and SL = 60 mol% (referred to as $S_{153}MB_{313}(60)$), which showed the best actuation performance.

With an aim to improve the cation transport properties of Im-doped $S_{153}MB_{313}(60)$, various types of ionic additives were introduced in a 1:1 molar ratio (additive: Im in polymer). Three different zwitterions were synthesized by tethering a sulfonate anion and different quaternary cations, that is, 3-(1-methyl-3-imidazolium) propanesulfonate (hereafter, ZImS), 3-[ethyl (dimethyl)ammonio]-1-propanesulfonate (ZAmS) and 3-(1-methylpiperidinium)-1-propanesulfonate (ZPiS). The chemical structures of ZImS, ZAmS and ZPiS are shown in Fig. 1b. The same sulfonate anion was used in the zwitterions in order to achieve good thermodynamic compatibility with the PSS chains, whereas the cation was varied to control the degree of intermolecular interactions occurring in the PSS phases. A conventional ionic liquid comprising the Im cation and bis(trifluoromethane sulfonyl)imide (TFSI$^-$) anion (hereafter, referred to as Im/TFSI) was also employed as the control.

The strength of the ion pairs as well as the dipole moment of the ionic complexes that could be formed in the PSS phases were estimated by *ab initio* calculations using a density functional theory based on the B3LYP exchange-correlation functional. The 0-K binding energies of ion pairs are listed in Table 1, for which the geometry had been optimized. The Im cation was expected to bind more strongly to the polymer matrix than to the zwitterion, despite the presence of the same sulfonate anion. We found noticeable reductions in the strength of ionic interactions between the zwitterion and the PSS, regardless of the type of cation in the zwitterion, which suggests the presence of more attractive interactions between the PSS and the Im cation than between the PSS and the zwitterions. The significantly high dipole moment formed by the Im/PSS/ZImS interactions is also noteworthy.

### Performance of actuators based on cation-conducting polymers.
We fabricated actuators in a simple bimorph configuration by sandwiching cation-conducting polymers between single-wall carbon nanotube (SWCNT) electrodes, as schematically illustrated in Fig. 2a. The dimension of the actuator strips was 2 mm × 11 mm × 80 µm. Supplementary Fig. 1 shows a typical cross-sectional scanning electron microscope (SEM) image of the actuator. The hexagonal cylindrical (HEX) morphology was observed for all polymers with and without ionic additives.

**Table 1 | *Ab initio* interaction energies of a range of ionic interactions and dipole moments for different types of ionic complexes at 0 K in a vacuum.**

| Type of interaction | Binding energy (kJ mol$^{-1}$)* | Type of ionic complex | Dipole moment (Debye)* |
|---|---|---|---|
| *Im/polymer* | | | |
| Im/C$_6$H$_5$SO$_3^-$ | 459.3 | Im/C$_6$H$_5$SO$_3^-$/ZImS | 12.66 |
| *Im/zwitterion* | | | |
| Im/ZImS | 307.0 | | |
| Im/ZAmS | 299.7 | Im/C$_6$H$_5$SO$_3^-$/ZAmS | 7.30 |
| Im/ZPiS | 321.6 | | |
| *Zwitterion/polymer* | | | |
| ZImS/C$_6$H$_5$SO$_3^-$ | 229.9 | | |
| ZAmS/C$_6$H$_5$SO$_3^-$ | 227.4 | Im/C$_6$H$_5$SO$_3^-$/ZPiS | 7.46 |
| ZPiS/C$_6$H$_5$SO$_3$ | 217.2 | | |

ZAmS, 3-[ethyl(dimethyl)ammonio]-1-propanesulfonate; ZImS, 3-(1-methyl-3-imidazolium) propanesulfonate; ZPiS, 3-(1-methylpiperidinium)-1-propanesulfonate.
*All calculations were performed using a density functional theory (DFT) exchange-correlation functional, B3LYP.

Transmission electron microscope (TEM) image in Fig. 2a displays the representative HEX morphology of the Im-doped S$_{153}$MB$_{313}$(60) containing ZImS, where the PSS phases were darkened by RuO$_4$ staining. The TEM image also indicates that the ion migration in the polymer layer occurs across a major PSS phases and, therefore, the issue of orientation of the micro-domains can be attenuated.

It should be noted here that, although analogous HEX morphology was determined for all samples, the degree of ordering and domain size were largely deviated. In Supplementary Fig. 2, we show small-angle X-ray scattering (SAXS) profiles and the plots of domain sizes for Im-doped S$_{153}$MB$_{313}$(60) with zwitterions (or Im/TFSI), compared with the results obtained without additives. The best long-range ordering and the largest increment in domain size of *ca.* 18% were obtained with the addition of ZImS, in contrast to the relatively poor ordering seen with other additives. This should be intimately related to the improved thermodynamic compatibility of ZImS with Im-doped PSS phases to yield enhanced segregation strength of microdomains.

Figure 2b shows the time-dependent bending response of the actuators at alternating square-wave voltages of $\pm 3$ V with a cycle of 20 s, at room temperature and under ambient laboratory atmosphere conditions. The displacement ($\delta$) was measured from the position of the actuator tip and a negative sign in the $\delta$ value indicates that $-3$ V was applied to the actuator to yield the bending motion in the opposite direction. The strain ($\varepsilon$) was calculated based on the $\delta$ values[25].

The $\delta$ value of the actuator based on the Im-doped S$_{153}$MB$_{313}$(60) was as low as 4.5 mm (corresponding to strain, $\varepsilon = 0.6\%$) and $\geq 20$ s was needed to reach equilibrium displacement, which was attributed to the limited Im cation diffusion in the polymer layer. When the Im/TFSI was added to the Im-doped S$_{153}$MB$_{313}$(60), the actuator showed more than 40% improvement in the $\delta$ value. An increased amount of ions and the plasticizing effects of the TFSI$^-$ anion (lowering the glass transition temperature of PSS) are responsible for the improved bending motion. However, the response time of the actuator remained slow, consistent with that observed in studies on ionic polymer actuators containing ionic liquids[5,8,25].

Furthermore, we observed that the performance of ionic polymer actuators based on Im-doped S$_{153}$MB$_{313}$(60) can be largely modulated by the addition of zwitterions. In particular, the largest $\delta$ value exceeding 14 mm (corresponding to $\varepsilon = 1.8\%$) with a noticeable reduction in the response time was achieved for the actuator containing ZImS. The actuator also displayed a fast restoration motion at a negative potential, which suggests the occurrence of reversible and fast ion migration under the given stimuli. Photographs shown in the bottom panel of Fig. 2b were taken at 1 s after applying $\pm 3$ V to illustrate the fast bending motion of the actuator containing ZImS. The other actuators containing ZAmS and ZPiS showed relatively low $\delta$ values of 7 and 4 mm, respectively, which was indicative of the importance of the zwitterion type in improving actuation performance.

The dissimilar displacements with zwitterions were believed to be related to the ionic conductivities of the polymer layers. As shown in Fig. 2c, we observed the lowest ionic conductivity for Im-doped S$_{153}$MB$_{313}$(60); this could be enhanced by several orders of magnitude by adding the ionic liquid, Im/TFSI, as the sample is capable of conducting both cations and anions. When the zwitterions were introduced into the Im-doped S$_{153}$MB$_{313}$(60), we observed a markedly high conductivity with ZImS that surpassed the conductivity of the Im/TFSI-embedded analogue. This is intriguing because ZImS itself does not contribute to ionic conductivity, owing to the electroneutrality. In contrast, the effects of ZAmS and ZPiS on conductivity were insignificant, responsible for minor improvements in electro-mechanical deformation (Fig. 2b). By Vogel–Tammann–Fulcher analysis we obtained dissimilar activation barrier for ion conduction. Solid lines in Fig. 2c represent the analysis using the Vogel–Tammann–Fulcher equation, yielding dissimilar potential barriers to ion conduction of 873, 1,249, 1,552 and 920 K for samples containing ZImS, ZAmS, ZPiS and Im/TFSI, respectively.

In order to determine the effects of zwitterions on the dielectric response of the polymer layer, as shown in Fig. 2d, the static dielectric constant ($\varepsilon_s$) was evaluated for each sample, as shown by fitted lines in the figure[26]. The Im-doped S$_{153}$MB$_{313}$(60) exhibited the lowest $\varepsilon_s$ of 7, which could be increased by introducing ionic additives. Notably, the $\varepsilon_s$ value radically increased to 76 with ZImS, indicating that the Im cation in the polymer containing ZImS can readily dissociate or reform, subsequently leading to a large increase in charge density. We infer the presence of synergistic dipole alignment in the zwitterion-containing PSS phases while the dipoles in conventional ionic liquid prefer antiparallel alignment, lowering the $\varepsilon_s$ value[27].

These results suggest that the mechanism underlying the marked improvements in bending strain and response time in the actuators containing ZImS involves facilitated ion dissociation in the high dielectric constant matrix along with fast Im cation migration through well-defined ionic phases.

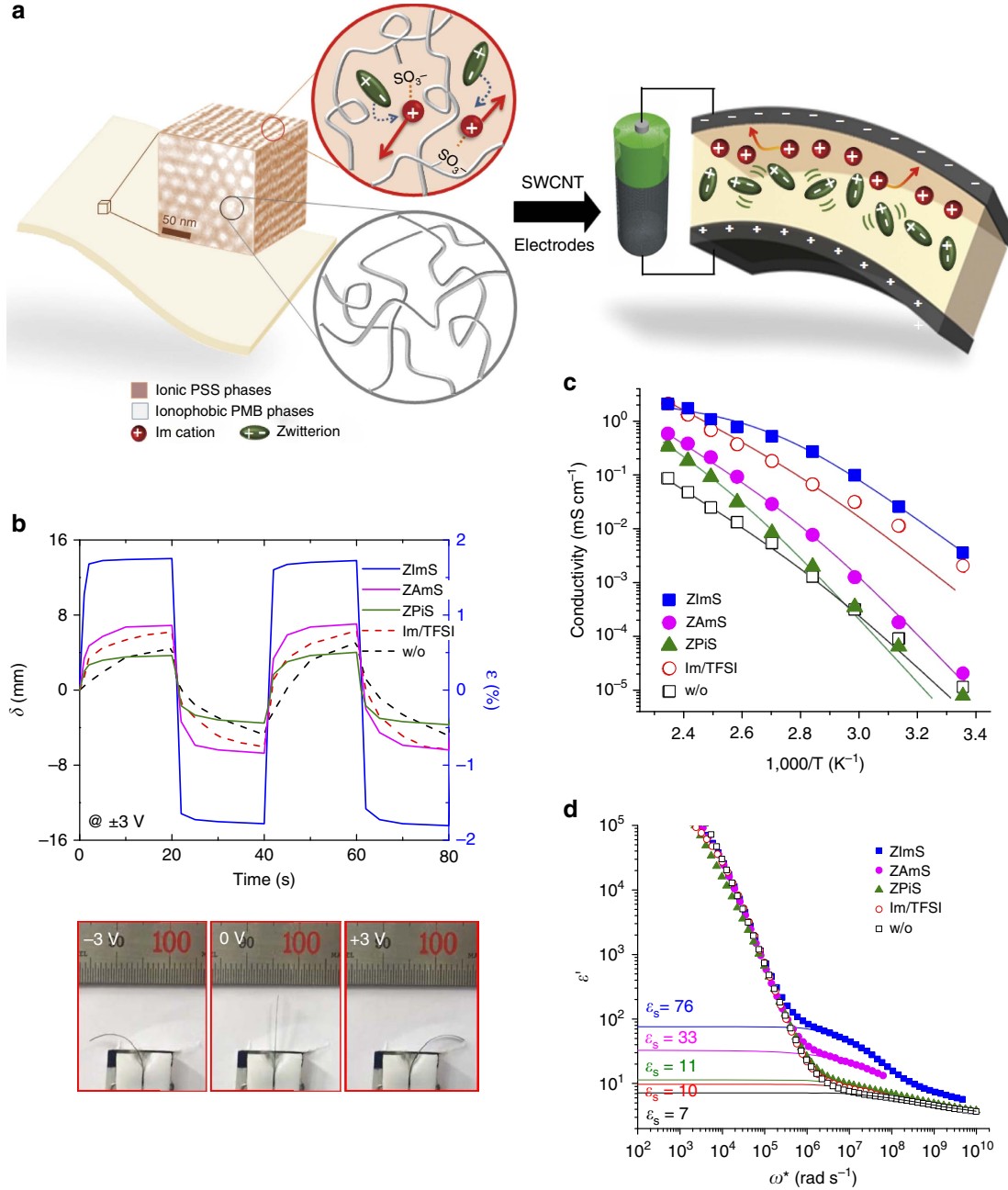

**Figure 2 | Electromechanical properties of actuators.** (**a**) Schematic illustration of the actuators comprising cation-conducting block copolymer sandwiched between SWCNT electrodes. A cross-sectional TEM image obtained from Im-doped $S_{153}MB_{313}(60)$ containing ZImS was used to draw the schemes, which shows the [001] view of HEX morphology where the PSS phases were darkened by $RuO_4$ staining. (**b**) Displacement ($\delta$) and bending strain ($\varepsilon$) of the actuators comprising Im-doped $S_{153}MB_{313}(60)$ and ionic additives at alternating square-wave voltages of $\pm 3\,V$ and a frequency of 0.025 Hz. Photographs in the bottom panel were obtained from the actuator containing ZImS. (**c**) Conductivities and (**d**) dielectric permittivity spectra of Im-doped $S_{153}MB_{313}(60)$ with various ionic additives, compared with those without additive. Solid lines in **c** represent the analysis using the Vogel–Tammann–Fulcher (VTF) equation. The static dielectric constants, $\varepsilon_s$, are shown by fitted solid lines in **d**.

We note here that for single-ion electroactive actuators, the bending curvature ($\mathbb{K}$) and electro-actuation number ($\alpha$) can be expressed as equation (1)[21].

$$\mathbb{K} = \alpha\sqrt{1-\gamma}\,\frac{V}{\sqrt{\frac{V}{8}+1}}, \quad \alpha = \frac{3\,K\,S\,v}{4\sqrt{\pi}E\,S_0\,h}\sqrt{\frac{c_0}{l_B}} \quad (1)$$

where $\gamma$ is ion-crowding parameter, $V$ is applied voltage, $K$ is bulk modulus of polymer, $E$ is Young's modulus of actuator, $S$ is contact area of electrode, $S_0$ is projected area of electrode, $v$ is

excess volume associated with cation, $h$ is half-thickness of polymer layer, $c_0$ is number density of cation and $l_B$ is Bjerrum length.

This gives the expression for bending curvature in relation with dielectric constant ($\varepsilon$)

$$\mathbb{K} \propto \sqrt{\varepsilon} \quad (2)$$

The response time of the actuator can be related to the relaxation time scale ($\tau$)[21].

$$\tau = R_0 C_0; \quad R_0 = \frac{2\,h}{S_0\,\sigma}, \quad C_0 = \frac{\varepsilon\,S}{8\,\pi\,l_D} \quad (3)$$

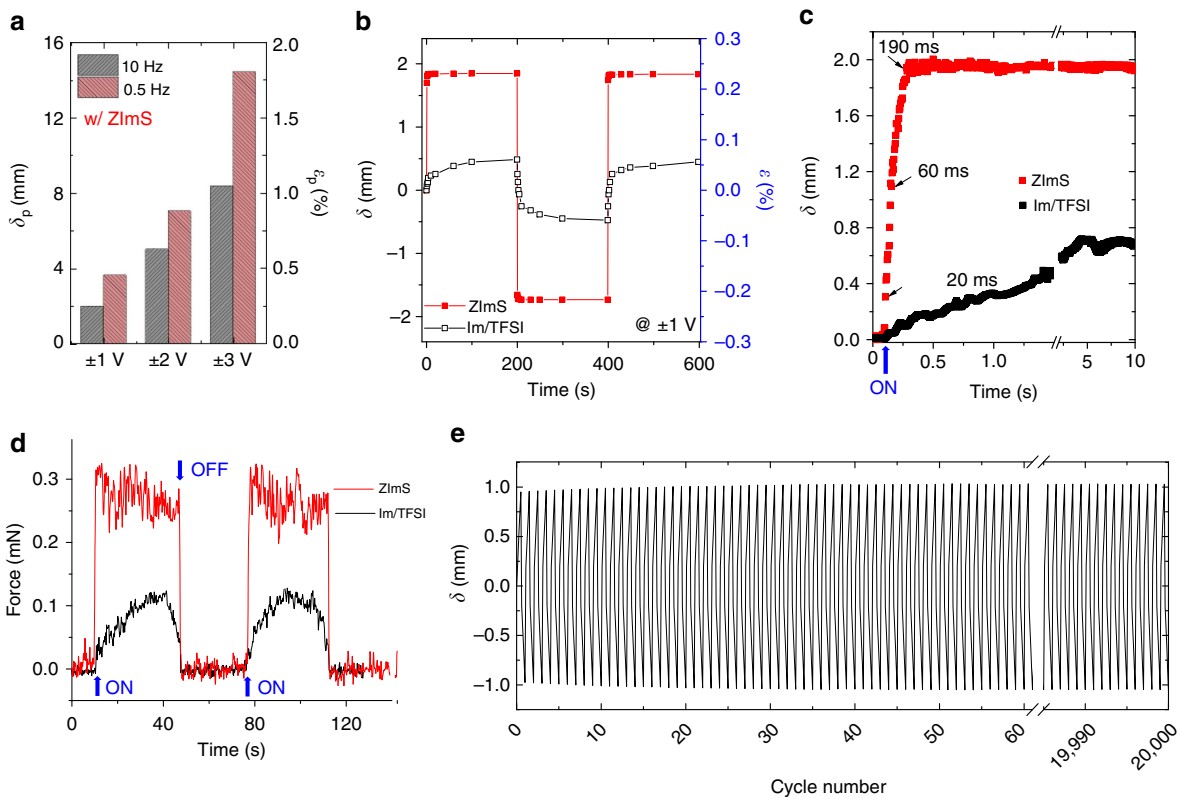

**Figure 3 | Low-voltage-driven superfast actuators.** (**a**) Peak-to-peak displacement ($\delta_p$) and bending strain ($\varepsilon_p$) of the actuator containing ZImS at voltages of ±1, ±2 and ±3 V and frequencies of 0.5 and 10 Hz. (**b**) The $\delta$ and $\varepsilon$ values of the actuator containing ZImS at alternating square-wave voltages of ±1 V, in comparison with the actuator containing Im/TFSI. (**c**) Laser-displacement measurements and (**d**) the force generation of the actuator with ZImS, compared with that containing Im/TFSI, by applying 1 V. (**e**) Cycle life of the actuator under continuous operation in air at ±1 V and 10 Hz.

where $\tau$ is RC time constant, $R_0$ is resistance of the bulk, $C_0$ is linear Debye capacitance, $\sigma$ is ionic conductivity of polymer and $l_D$ is Debye length.

We obtain the $\tau$

$$\tau \propto \frac{\sqrt{\varepsilon}}{\sigma} \qquad (4)$$

Equations (2) and (4) indicate that the actuator containing ZImS would reveal *ca.* three times larger bending displacement and two orders of magnitudes faster response time than the actuator without additive, in good agreement with our experimental data.

Hereafter, we focus on the results of the best-performing actuator containing ZImS, and compare this actuator to the Im/TFSI-embedded analogue as the control. We assessed the single-ion-conducting properties of the Im-doped $S_{153}MB_{313}(60)$ containing ZImS, compared with that of the Im/TFSI-containing analogue, by polarization experiments[28]. Supplementary Fig. 3 displays current density profile of each sample after polarization. The ratio of the current flow at steady state to the initial current flow ($I_{ss}/I_0$) was determined to be 0.88 for Im-doped $S_{153}MB_{313}(60)$ containing ZImS, which far exceeds the low value of 0.58 of Im/TFSI-embedded analogue.

**Superfast ionic polymer actuators containing zwitterions.** As the development of fast-response low-voltage-driven actuators is an aim of the present study, we explored the actuation performance of the actuators under low-voltage conditions. This approach was motivated by the fact that most ionic polymer actuators commonly display a large reduction in the generated strain following a decrease in the voltage of the activation field, an increase in frequency or both, which hampers their use in practical applications[8,11,25].

Figure 3a shows the peak-to-peak displacement ($\delta_p$) and bending strain ($\varepsilon_p$) of the actuator, which were measured with alternating square-wave voltages at ±1, ±2 and ±3 V and at frequencies of 0.5 and 10 Hz. For the actuator containing ZImS, the degree of electromechanical deformation was roughly proportional to the applied voltage. Most importantly, a large $\delta$ of 2.0 mm was readily achieved at a frequency of 10 Hz (a cycle time of 50 ms) at ±1 V, which markedly exceeded the best performance of other ionic polymer actuators reported thus far. The small difference in the bending strains with frequencies of 0.5 and 10 Hz under the low-voltage conditions is particularly noteworthy.

The low-voltage-driven actuation performance of the actuator based on the zwitterion-containing single-ion-conducting polymer was directly compared with that of actuators comprising conventional ionic liquid. As shown in Fig. 3b, a more than threefold improvement in the bending strain was demonstrated for the actuator with a superfast response time under ±1 V conditions, as compared with the findings of the actuator based on ionic liquid. Moreover, it should be noted that non-detectable back relaxation behaviour was observed with the actuator, with an extended interval of 200 s. These findings suggest that ion migration and accumulation occur in the actuator without any reverse migration. The measured current at applied voltages of ±1 V is shown in Supplementary Fig. 4, confirming both a full charging of the actuators and a very small leakage current.

For the accurate measurements of the response time of the actuator under low-voltage operating conditions, laser-displacement measurements were conducted. As shown in

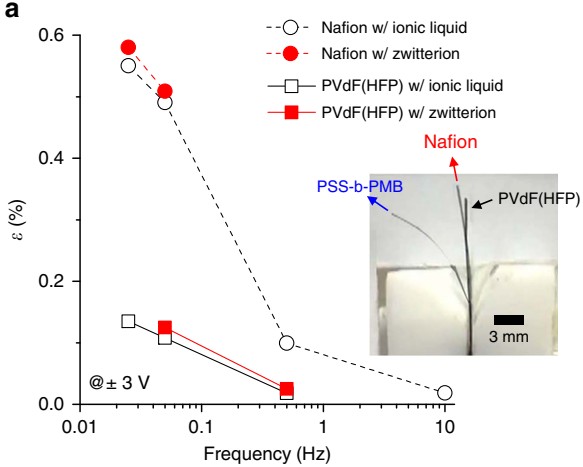

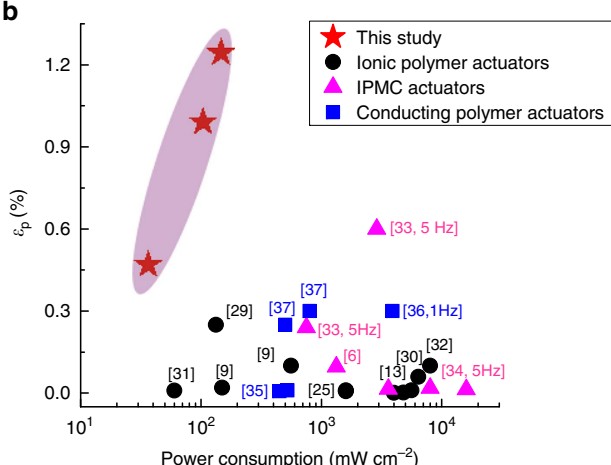

**Figure 4 | Low-power and high-performance actuators.** (**a**) Strain-frequency dependency of the actuators based on PVdF(HFP) and Nafion containing ZImS and Im/TFSI, measured with alternating square-wave voltages of $\pm 3\,V$. Photographs in the inset were taken at $-3\,V$ and $0.5\,Hz$ to directly compare the actuation performance of the actuator to that of actuators based on conventional polymers. (**b**) Strain-power consumption dependency of the actuator, compared with other types of soft actuators reported thus far, which clearly represents the impact of our results on the structure of widely studied actuators fabricated with polymeric materials.

Fig. 3c, after applying a voltage of 1 V, we found that the initial time to respond to the actuation field was as short as 20 ms (the arrow indicates when the voltage was turned on). In fact, the actuator readily moved 1 mm within 60 ms, and the movement was gradually suspended at 190 ms. This is tremendous progress as compared with the actuator with Im/TFSI-containing polymer, which require ~5 s to reach the equilibrium-bending motion. To our knowledge, this level of low-voltage-driven actuation performance has never been reported. In Supplementary Movie 1, we have provided a video demonstrating the successful use of the actuator as artificial fingers, which validates its use as bendable fingers at $\pm 1\,V$ and a frequency of 10 Hz.

As shown in Fig. 3d, the actuator can generate forces of ~0.3 mN within 0.5 s by applying 1 V; this force is threefold higher and the response time is several orders of magnitude faster than that of the Im/TFSI-containing sample under the same conditions. These values were recorded at the same free length of 11 mm. Given that the generated forces of most ionic polymer actuators reportedly range from 0.05 to 0.1 Hz, the high frequency > 1 Hz obtained with the actuator appears promising.

In addition to their large bending strains and superfast switching response, the actuators also have long-term durability, which is an important requirement for practical applications. The actuator displacement was monitored for extended times while applying $\pm 1\,V$ square-wave input signals with a cycle time of 50 ms. We found that the bending motion of the actuator was maintained over 20,000 cycles in air with negligible changes in the actuator stroke, as shown in Fig. 3e. This confirms that the rapid migration of Im cation across the polymer layer while dipole relaxation of zwitterions does not have a negative influence on the durability, thus enabling accurate and reliable motion of the actuator. Furthermore, upon examining the bending performance of the actuators over extended time periods, nonsignificant change in the bending strain was detected for 3 months, as shown in Supplementary Fig. 5. This confirms the negligible leaching of ionic additives from the actuators owing to the non-excessive amounts of Ims and zwitterions in the polymer layers.

**Soft single-ion-conducting actuators for micropower devices.** The performance of actuator corresponds to several times larger bending strain and over 100 times faster response time, compared with the actuators based on conventional PVdF(HFP) random copolymer (Kynar Flex 2801) and the state-of-the-art perfluoroacid ionomer, Nafion (Dupont) under the same conditions, as shown in Fig. 4a. This is remarkable as the PVdF(HFP) and Nafion have been the most widely employed polymers for ionic polymer actuators. Such difference becomes more evident by reducing operation voltage, where the Nafion actuators displayed negligible bending motion with strain of ~0.1% and slow response time over 10 s at $\pm 1\,V$ (detailed data are given in Supplementary Fig. 6). The bending strain could be slightly improved by adding ZImS; however, the addition of zwitterions was not found to reduce response time for both actuators based on PVdF(HFP) and Nafion.

This clearly implies that the role of the polymer matrix is to stimulate the fast bending response, where the requirements can be summarized as (1) is capable of single-ion conduction through tethered charges in the polymer backbone; (2) can confine ionic moieties into ionic channels with a high dielectric constant to facilitate effective ion dissociation and to lower the activation barrier for ion migration; and (3) should possess well-defined ionic domains to create less tortuous ion-conduction pathways. By designing multiblock copolymers, it is expected that the actuation performance can be further improved with high stress/strain rate, although the achievement of well-defined morphologies from multiblock copolymers containing ionic additives remains a work in progress. The employment of co-continuous morphologies or cubic structures having major conducting phases would be a means to secure more advanced actuators with good mechanical durability and high ionic conductivity[17]. Experiments to see whether the morphology effects are present in our actuators will also be a subject of our future studies.

The successful operation of the actuator at 1 V with a millisecond-scale response time emphasizes its use as a micropower device. The low-voltage-driven soft actuators currently being developed include ionic polymer actuators, ionic polymer metal nanocomposites actuators and conducting polymer actuators. By comparing the power consumption and generated strain of the actuator against these other types of actuators reported thus far, we found that the actuator represented significant progress as compared with previous technologies while consuming only 1/10th of the power in comparison (Fig. 4b). The literature values were collected from independent references published during the past 13 years[6,9,13,25,29–37]. This

confirmed that the actuators based on nanostructured single-ion conductors enable the successful actuator operation with a micropower consumption, thus paving the way for more advanced biomimetic technologies in the future.

In conclusion, we investigated actuators based on single-ion-conducting block copolymers through the introduction of zwitterions that offered a polar medium close to water, and accordingly increased the charge density and ionic conductivity. The actuator showed a large displacement in millimetre scales at 50 ms after applying 1 V and no back relaxation was detected. Results of the direct comparison of the actuator with the other actuators reported in the literature emphasize its massive potential in the field of soft robotics, artificial muscles and biomedical microdevices, given its micropower consumption.

## Methods

**Synthesis of sulfonated block copolymers.** Poly(styrene-b-methylbutylene) (PS-b-PMB) block copolymers were synthesized by sequential anionic polymerization of styrene and isoprene monomers using cyclohexane as a solvent and sec-butyl lithium as an initiator at 45 °C, followed by selective hydrogenation of isoprene units in the presence of a homogeneous Ni-Al catalyst in cyclohexane, using a 2L Parr batch reactor at 80 °C and 420 p.s.i. Molecular weights of PS-b-PMB block copolymers were determined using gel permeation chromatography (Waters Breeze 2 HPLC) in tetrahydrofuran (THF) with a light-scattering detector, where the $dn/dc$ value of PS-b-PMB was calculated based on block compositions. The polydispersity indices of the PS-b-PMB copolymers were less than 1.06. The PS chains of the PS-b-PMB block copolymers were partially sulfonated using acetic sulfate in 1,2-dichloroethane at 40 °C under a $N_2$ blanket to yield poly(styrenesulfonate-b-methylbutylene) (PSS-b-PMB) block copolymers. Samples with different SLs were prepared by controlling reaction time, as determined by $^1H$ nuclear magnetic resonance ($^1H$-NMR, Bruker AVB-300) spectroscopy in acetone-d6.

**Synthesis of zwitterions.** ZImS was synthesized by a one-step reaction of 1-methylimidazole and 1,3-propanesultone in acetone at room temperature for 5 days. ZAmS was synthesized by aminolysis of 1,3-propanesultone by dimethylethylamine in 1,2-dichloroethane at room temperature for 2 days. ZPiS was synthesized by 1-methylpiperidine and 1,3-propanesultone in dichloromethane at room temperature for 4 days. The resultant products were purified by filtering and precipitation to obtain a white powder (ZImS and ZAmS) and a pale pink power (ZPiS), followed by vacuum drying at 60 °C. The successful synthesis of zwitterions was confirmed by $^1H$-NMR and Fourier transform infrared (a Spectrum Two, PerkinElmer, USA) spectroscopy experiments. Im/TFSI was synthesized by mixing Im (≥99.5% HPLC grade) and bis(trifluoromethane sulfonyl)imide (≥97% HPLC grade) in equimolar ratio, followed by heating above the melting temperature of the ionic liquid.

**Preparation of polymer membranes containing ionic additives.** Inhibitor-free anhydrous THF (≥99.9%) was used without further purification, and methanol was degassed three times before use. Predetermined quantities of Im and PSS-b-PMB were weighed into glass vials, and 5 wt% solutions were prepared using methanol, followed by solvent-casting and vacuum drying at room temperature. Im-doped polymers were then re-dissolved into 80/20 vol.% methanol/THF mixtures and predetermined quantities of zwitterions (or ionic liquid) were added into the mixtures. Approximately 80-µm-thick membranes were prepared by solvent-casting on an aluminium mold (an area of 1 cm × 1.5 cm) at room temperature under an argon atmosphere for 2 days, followed by vacuum drying at 70 °C for 7 days. The dried polymer membranes were further subjected to a normal stress of ~200 kgf cm$^{-2}$ in a hydraulic Carver press at 25 °C for 1 h. As control samples, PVdF(HFP) membranes were prepared by the same procedures mentioned above. For the preparation of Nafion membranes, predetermined quantities of ionic additives were added into 10 wt% Nafion in $H_2O$ (Sigma-Aldrich) followed by solvent-casting. All sample preparation was performed in an Ar-filled glove box with oxygen and moisture concentration below 0.1 p.p.m.

**Morphology of polymer membranes containing ionic additives.** The morphology of polymer membranes prepared by aforementioned procedures was investigated by synchrotron SAXS experiments using the PLS-II 4C SAXS beam-line at the Pohang Accelerator Laboratory. The wavelength ($\lambda$) of the incident X-ray beam was 0.15 nm ($\Delta\lambda/\lambda = 10^{-4}$) and sample-to-detector distance of 2.0 m was used. The resulting two-dimensional scattering data were averaged azimuthally to obtain intensity versus scattering wave vector $q$ ($q = 4\pi\sin(\theta/2)/\lambda$, where $\theta$ is the scattering angle). The SAXS results were confirmed by cross-sectional TEM (Hitach H-800) imaging on cryo-microtomed specimens (RMC Boeckeler PT XL Ultramicrotome).

**Conductivity measurements.** In an Ar-filled glove box, the through-plane conductivity of polymer membranes was measured using an AC impedance spectroscopy (VersaSTAT3, Princeton Applied Research, AMETEK Inc.) using a home-built two-electrode cell with 1.25 cm × 1.25 cm stainless steel-blocking electrodes, Kapton spacers and 1 cm × 1 cm Pt working/counter electrodes. Data were collected over a frequency range of 5–50,000 Hz.

**Dielectric relaxation spectroscopy.** The dielectric measurements of polymer membranes were performed using dielectric relaxation spectroscopy (Novocontrol GmbH Concept 40 broadband dielectric spectrometer). The polymer membranes were sandwiched by two polished brass electrodes (10 mm-diameter top electrode and 30 mm-diameter bottom electrode) and a gap was controlled by a silica spacer. Samples were annealed in the Novocontrol sample chamber at 170 °C for 2 h before measurement to remove any moisture and to make good electrolyte/electrode contact. The dielectric permittivity was measured using a sinusoidal voltage with amplitude 0.1 V in a frequency range of $10^{-1}$–$10^7$ Hz.

**Potentiostatic polarization experiments.** Potentiostatic polarization experiments (VersaSTAT3, Princeton Applied Research, AMETEK Inc.) were conducted using hermetically sealed coin cells (CR2032, MTI Corporation). The polymer membranes were sandwiched between Al-foils (MTI Corporation) and annealed at 60 °C for 24 h before the measurements at room temperature. Cells were polarized using potentials, $\Delta V$, of 50 mV for all samples. The ac impedance spectroscopy measurements were performed before and after the polarization.

**Preparation of actuators.** SWCNTs (produced by a CoMoCAT catalytic chemical vapor deposition (CVD) process) were purchased from Sigma-Aldrich and used without further purification to prepare SWCNT electrodes. The electrodes contain [EMIm][BF$_4$], PVdF(HFP) (Kynar Flex 2801, Arkema Chemical Inc.) and SWCNTs in a weight ratio of 2.5:1.5:1.0. The thickness of the electrode prepared by solvent-casting was ca. 10 µm. The polymer membranes were then sandwiched by two SWCNT electrodes via hot pressing to yield a trilaminar structure. The dimension of actuators employed in the present study was 13 mm × 1 mm × 80 µm. The cross-sectional structure of the actuators was examined by field emission scanning electron microscope (FE-SEM, Phillips electron optics B. V., XL30S FEG) with a 5 keV accelerating voltage.

**Actuator performance tests.** The actuator strips were clipped with two Pt disk electrodes yielding 11 mm free length from Pt contacts. The bending motion of the actuators was monitored by applying square-wave voltages to the actuator strips using a potentiostat (VersaSTAT3, Princeton Applied Research, AMETEK Inc.) under ambient laboratory atmosphere with an average relative humidity of 25 ± 5%. For the accurate measurements of response time, the laser-displacement sensor (Gocator2330, LMI Technologies) was placed that the laser beam was perpendicular to the surface of the actuator. The displacement signal was monitored using a time interval of 5 ms (the laser beam with a scan rate of 200 Hz). The displacement ($\delta$) of the actuator was transformed into the bending strain ($\varepsilon$) based on $\varepsilon = 2\delta d/(L^2 + \delta)$: $L$ is the free length from Pt contacts and $d$ is the thickness of actuator.

**Blocking force measurements.** The blocking forces of the actuators were measured using nano universal testing machine (Nano UTM, MTS Nano Instruments, USA) equipped with a load cell with a maximum load of 500 mN and a resolution of 50 nN. A dc step voltage was applied to the actuators for 30 s using a dc power supply (DRP-303D, Digital electronics), and the load on specimen was recorded using the TestWorks 4 software. All actuation tests were carried out under ambient laboratory atmosphere with an average relative humidity of 25 ± 5%.

**Data availability.** The data that support the findings of this study are available from the corresponding author on request.

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

## Acknowledgements

This work was supported by the Samsung Research Funding Center of Samsung Electronics under Project Number SRFC-MA1402-08.

## Author contributions

M.J.P. conceived the idea and designed the project and O.K. carried out the experiments and analysed data. O.K. and H.K. carried out the actuation performance tests, and U.H.C. carried out the dielectric spectroscopy experiments. M.J.P. and O.K. wrote the paper.

## Additional information

**Competing financial interests:** The authors declare no competing financial interests.

