## [Peer Review File · Nature Communications]

Reviewer #1 (Remarks to the Author)

This manuscript entitled as "1 Volt-Driven, Superfast Soft Actuator: Single Ion Conductors Powering Artificial Muscles" reports effects of zwitterions in ion-conducting polymers on performance and response time. This manuscript is well organized and novel enough to be published on Nature Communications. However, there are several points and questions before the publication.

1) In the introduction part, the authors emphasized a requirement of fast response like piezoelectric and dielectric response. However, the reported values on speed of piezoelectric and dielectric response under DC signal are around tens of nanoseconds, which ionic conducting polymer actuator has not reached. I think it would be better to mention the other quantitative goals such as a response time in the introduction part.

2) The authors pointed out that an increase of dielectric constant with zwitterions as the main reason for enhancement of performance and response time. However, the quantitative explanation for effects of dielectric constant onto actuator performances is missing in this manuscript.

Reviewer #2 (Remarks to the Author)

This manuscript represents a continued discourse in the literature for the formation of electromechanical transducers with exceptional deflection rates for potential biomimetic applications. This field has remained active for over the past decade with a combination of mechanical engineers and chemists, working in parallel to address this key issue. The key results are the formation of block copolymers as charged templates for the introduction of zwitterionic liquids resulting in the observation of some of the fastest rates of deflection reported to date with excellent device stability. The manuscript is grounded in solid polymer morphological analysis, building on the well established expertise of this author and his team. The report is very significant, especially as this field was looking for this advance to rejuvenate this field across the globe.

The originality of this manuscript stems from a unique combination of the earlier literature. Other authors have reported the use of sulfonated block copolymers and neutralization with imidazoles to form imidazoliums as electromechanical transducers. The synthesis of the block copolymers, post polymerization hydrogenation, and sulfonation are well established synthetic methods in the literature. Zwitterionic ionic liquids are also an active area of research for the field of ionic liquids, and other researchers have also reported zwitterionic polymers due to the high dielectric constant and unique aggregation phenomena. However, this manuscript represents the first example of the combination of zwitterionic ionic liquids in combination with a single ion polymeric conductor for the fabrication of an electromechanical transducer.

The quality of the presentation is excellent, the observations are well supported with the expected experimentation and also the addition of supporting morphological information. This work builds nicely on the earlier work of Winey, Elabd, Lodge, and others for the investigation of tailored morphologies on the electromechanical response. The statistical treatment of the data is acceptable, and the observations are well beyond the error of the measurements.

The referencing is comprehensive and represents leaders in the field, leveraging existing observations and continuing the literature discourse using accepted experiments.

Overall, the manuscript represents a significant advance in the field. There was some concern for the design of the diblock copolymer rather than building on recent advances using triblock copolymers and other multiblock copolymers. The authors should discuss more thoroughly the selection of the diblock architecture and its relevance to the optimization of performance. In addition, the authors should discuss more clearly the role of the hexagonal morphology and the

tuning of the phase diagram as a function of the addition of ionic liquid. What is the expected optimal morphology, a co-continuous morphology should be discussed as a means to maintain mechanical durability and ion conductivity? There was also some modest concern for the inclusion of the "artificial muscles" term in the title, since the manuscript did not demonstrate any physiological conditions or model conditions that would give a stronger vision for in-vivo applications. The authors should also comment on any leaching of the ionic liquids from the membranes as a function of time and a function of IL incorporation.

Response to Reviewer # 1

Ms No.: NCOMMS-16-18133

Title: “1 Volt-Driven, Superfast Soft Actuators: Single Ion Conductors Powering Artificial Muscles”

Authors: Onnuri Kim, Hoon Kim, U Hyeok Choi, Moon Jeong Park*

First of all, we would like to thank the reviewer for a very thorough reading of the manuscript and many helpful comments. We have made changes to the manuscript in response, as detailed below.

General comments: This manuscript is well organized and novel enough to be published on Nature Communications.

Response: We thank the reviewer for the encouraging words.

Question 1: In the introduction part, the authors emphasized a requirement of fast response like piezoelectric and dielectric response. However, the reported values on speed of piezoelectric and dielectric response under DC signal are around tens of nanoseconds, which ionic conducting polymer actuator has not reached. I think it would be better to mention the other quantitative goals such as a response time in the introduction part.

Answer 1: To reflect the reviewer’s comment, in Introduction of the revised manuscript, we stated the quantitative goals of ionic polymer actuators in terms of response time and actuation field.

“If the ionic polymer actuators can respond quickly—in milliseconds like piezoelectric²³ and dielectric actuators²⁴—with a small battery (< 1.5V) and without sacrificing its bending strain, it could present a new chapter in the development of artificial muscles.”

Please note that commonly reported response times of piezoelectric actuators and dielectric actuators are in several milliseconds (sometimes in several tens of microseconds, but not nanoseconds).

Question 2: The authors pointed out that an increase of dielectric constant with zwitterions as the main reason for enhancement of performance and response time. However, the quantitative explanation for effects of dielectric constant onto actuator performances is missing in this manuscript.

Answer 2: As the reviewer suggested, we provided the expressions for bending curvature and response time in relation with dielectric constant in pages 11 ~ 12 of the revised manuscript. Thank you for prompting us to examine our paper more carefully.

“We note here that for single-ion electroactive actuators, the bending curvature (\mathcal{K}) and electro-actuation number (α) can be expressed as equation (1).²¹

$$\mathcal{K} = \alpha \sqrt{1-\gamma} \frac{V}{\sqrt{\frac{V}{8}+1}}, \quad \alpha = \frac{3 K S v}{4 \sqrt{\pi} E S_0 h} \sqrt{\frac{c_0}{l_B}} \quad (1)$$

where γ is ion crowding parameter, V is applied voltage, K is bulk modulus of polymer, E is Young's modulus of actuator, S is contact area of electrode, S_0 is projected area of electrode, v is excess volume associated with cation, h is half-thickness of polymer layer, c_0 is number density of cation, and l_B is Bjerrum length.

This gives the expression for bending curvature in relation with dielectric constant (ϵ)

$$\mathcal{K} \propto \sqrt{\epsilon} \quad (2)$$

The response time of the actuator can be related to the relaxation time scale (τ).²¹

$$\tau = R_0 C_0 \quad ; \quad R_0 = \frac{2 h}{S_0 \sigma}, \quad C_0 = \frac{\epsilon S}{8 \pi l_D} \quad (3)$$

where τ is RC time constant, R_0 is resistance of the bulk, C_0 is linear Debye capacitance, σ is ionic conductivity of polymer, and l_D is Debye length.

We obtain the τ

$$\tau \propto \frac{\sqrt{\epsilon}}{\sigma} \quad (4)$$

Equations (2) and (4) indicate that the actuator containing ZImS would reveal ca. 3 times larger bending displacement and two orders of magnitudes faster response time than the actuator without additive, in good agreement with our experimental data.”

Response to Reviewer # 2

Ms No.: NCOMMS-16-18133

Title: “1 Volt-Driven, Superfast Soft Actuators: Single Ion Conductors Powering Artificial Muscles”

Authors: Onnuri Kim, Hoon Kim, U Hyeok Choi, Moon Jeong Park*

First of all, we would like to thank the reviewer for a very thorough reading of the manuscript and many helpful comments and corrections. We have made changes to the manuscript in response, as detailed below.

General comments: The key results are the formation of block copolymers as charged templates for the introduction of zwitterionic liquids resulting in the observation of some of the fastest rates of deflection reported to date with excellent device stability. The manuscript is grounded in solid polymer morphological analysis, building on the well established expertise of this author and his team. The report is very significant, especially as this field was looking for this advance to rejuvenate this field across the globe.

Response: We thank the reviewer for the encouraging words.

Question 1: There was some concern for the design of the diblock copolymer rather than building on recent advances using triblock copolymers and other multiblock copolymers. The authors should discuss more thoroughly the selection of the diblock architecture and its relevance to the optimization of performance.

Answer 1: We agree with the reviewer’s point. As demonstrated by many research groups (Long, Koo, Spontak, Watanabe) and our own work (ref 14, “Factors affecting electromechanical properties of ionic polymer actuators based on ionic liquid-containing sulfonated block copolymers” *Macromolecules* **47**, 4357–4368 (2014)), the use of multiblock copolymers is expected to be beneficial in improving mechanical properties and thereby the actuation performance at high stress/strain rate. The employment of triblock copolymers in single-ion conducting actuators is currently underway; however, the development of high-performance actuators composed of triblock copolymers does not appear to be feasible at present. The major difficulty facing us is the lack of organization of zwitterion-containing sulfonated triblock copolymers. We hope that we will solve this problem by optimizing casting-solvents and annealing conditions.

To reflect the review's comment, we added a sentence regarding the perspective of multi-block architecture to improve actuation performance in page 16, lines 17 ~ 20 of the revised manuscript.

“By designing multi-block copolymers, it is expected that the actuation performance can be further improved with high stress/strain rate although the achievement of well-defined morphologies from multi-block copolymers containing ionic additives remains a work in progress.”

Question 2: In addition, the authors should discuss more clearly the role of the hexagonal morphology and the tuning of the phase diagram as a function of the addition of ionic liquid. What is the expected optimal morphology, a co-continuous morphology should be discussed as a means to maintain mechanical durability and ion conductivity?

Answer 2: Investigation of morphology effects on the performance of single-ion conducting polymer actuators is an important research topic for us. Based on our previous studies on morphology-conductivity relationship of ionic liquid-containing block copolymers, we are convinced that the achievement of cubic morphologies having major conducting phases is the desired direction to go; however, we have not yet obtained rich morphologies from sulfonated block copolymers containing zwitterionic liquids.

As the reviewer suggested, we added a brief discussion regarding the role of block copolymer morphology in improving actuation performance in page 16, lines 20 ~ 23 of the revised manuscript.

“The employment of co-continuous morphologies or cubic structures having major conducting phases would be a means to secure more advanced actuators with good mechanical durability and high ion conductivity.¹⁷ Experiments to see if the morphology effects are present in our actuators will also be a subject of our future studies.”

Question 3: There was also some modest concern for the inclusion of the "artificial muscles" term in the title, since the manuscript did not demonstrate any physiological conditions or model conditions that would give a stronger vision for in-vivo applications.

Answer 3: To reflect the reviewer's concern, we rephrased the title as “1 volt-driven, superfast polymer actuators: Single ion conductors powering soft robots”.

Question 4: The authors should also comment on any leaching of the ionic liquids from the membranes as a function of time and a function of IL incorporation.

Answer 4: We examined the time-dependent bending performance of our actuators at

applied voltages of ± 3 V. Representative results are shown below where the data were taken from the actuators containing ZImS. As can be seen from the figure, non-significant change in the displacement was detected for 3 months, signaling negligible leaching of ionic additives from the membranes. This is, in part, because of low contents of zwitterionic liquids (< 30 wt%) in our single-ion conducting polymers.

<Figure 1. The peak-to-peak bending strain of the actuator containing ZImS with extended time periods, measured at alternating square-wave voltages of ± 3 V and frequencies of 0.05 Hz and 0.025 Hz.>

To reflect the reviewer's point, we added a sentence regarding the leaching of ionic additives from the membranes in page 14, lines 17 ~ 21 of the revised manuscript. The plot of time-dependent bending strain of our actuators is also provided in Fig. S5 of the revised Supplementary Information.

"Furthermore, upon examining the bending performance of the new actuators over extended time periods, non-significant change in the bending strain was detected for 3 months, as shown in Fig. S5 of Supplementary Information. This confirms the negligible leaching of ionic additives from the actuators owing to the non-excessive amounts of imidazoles and zwitterions in the polymer layers."

Reviewer #1 (Remarks to the Author)

The manuscript has been well revised by reflecting the reviewers' comments. I recommend the publication of this manuscript to the Nature Communications.

Reviewer #2 (Remarks to the Author)

The revisions add to the scope and impact of this manuscript and also continue the various themes of discussion in the literature. The added sentences and revised titles more clearly position your paper for impact and citation. Well done.

The title remains a bit awkward, the title as proposed in your response to the reviewers is different from the title in your revised manuscript. I prefer the title in the response to the reviewers, can you change the title in your manuscript to match the title in your response.

This manuscript will have broad impact beyond only biomimetic devices, this title is more encompassing and more reflective of your work.